# Remodeling of lipid-foam prototissues by network-wide tension fluctuations induced by active particles

Andre A. Gu [1,2], Mehmet Can Uçar [3,11], Peter Tran [2], Arthur Prindle[2,4,5,6,7], Neha P. Kamat [1,5,8] & Jan Steinkühler [9,10] ✉

Recent advances in the field of bottom-up synthetic biology have led to the development of synthetic cells that mimic some features of real cells, such as division, protein synthesis, or DNA replication. Larger assemblies of synthetic cells may be used to form prototissues. However, existing prototissues are limited by their relatively small lateral dimensions or their lack of remodeling ability. Here, we introduce a lipid-based tissue mimetic that can be easily prepared and functionalized, consisting of a millimeter-sized "lipid-foam" with individual micrometer-sized compartments bound by lipid bilayers. We characterize the structural and mechanical properties of the lipid-foam tissue mimetic, and we demonstrate self-healing capabilities enabled by the fluidity of the lipid bilayers. Upon inclusion of bacteria in the tissue compartments, we observe that the tissue mimetic exhibits network-wide tension fluctuations driven by membrane tension generation by the swimming bacteria. Active tension fluctuations facilitate the fluidization and reorganization of the prototissue, providing a versatile platform for understanding and mimicking biological tissues.

Recent developments have led to the development of synthetic cells that are able to mimic some features of real cells[1], such as membrane fusion[2,3], division[4,5], and DNA replication[6]. Transferring these advances to larger, tissue-like scales is highly desirable. Conceptually, the construction of larger tissue mimetics demands the engineering of adhesive interactions between individual synthetic cells. In practice, achieving this is challenging: the interactions need to be strong enough for stabilization of tissue-scale structures, but they also need to allow for dynamic reorganization[7,8], a hallmark characteristic of biological tissues. For example, cross-linked populations of giant unilamellar vesicles were shown to assemble into millimeter-sized prototissues[9],

but these structures appeared static after assembly, indicating that they lack the fluidity of biological tissues. Advances have been made by the construction of proto-spheroids, where shape changes could be fueled by enzymatic reactions. However, these spheroids were sized only about 100 μm, indicating that the stabilizing interactions for the formation of larger tissue mimetics were too weak[10]. Similarly, printed lipid monolayer coated droplets immersed in an oil phase (droplet interface bilayers, DIBs) have been also suggested as a tissue mimetic[11–13]. Such droplet networks reach millimeter sizes and allow for osmotically induced reshaping. However osmotically induced forces are typically much larger than forces generated by molecular motors,

[1]Department of Biomedical Engineering, Northwestern University, Evanston, IL, USA. [2]Department of Chemical and Biological Engineering, McCormick School of Engineering, Northwestern University, Evanston, IL, USA. [3]Institute of Science and Technology Austria, Klosterneuburg, Austria. [4]Department of Biochemistry and Molecular Genetics, Feinberg School of Medicine, Northwestern University, Chicago, IL, USA. [5]Center for Synthetic Biology, Northwestern University, Evanston, IL, USA. [6]Department of Microbiology-Immunology, Feinberg School of Medicine, IL Chicago, USA. [7]Chan Zuckerberg Biohub Chicago, IL Chicago, USA. [8]Chemistry of Life Processes Institute, Northwestern University, Evanston, IL, USA. [9]Bio-Inspired Computation, Institute of Electrical and Information Engineering, Kiel University, Kiel, Germany. [10]Kiel Nano, Surface and Interface Science KiNSIS, Kiel University, Kiel, Germany. [11]Present address: School of Mathematical and Physical Sciences, University of Sheffield, Sheffield, UK. ✉e-mail: jst@tf.uni-kiel.de

the latter of which should be sufficient to remodel bio-inspired tissues. Additionally, the surrounding oil phase limits emulsion-based tissue and DIB prototissues in their biocompatibility. Most recently, DIBs were transferred from the oil phase to the water phase, but the resulting structures were either limited in their size or demonstrated no remodeling abilities[14–16]. Another model explored as a tissue mimetic involves oil-in-water droplets, where adhesion strength is precisely tuned to allow tissue-like remodeling[8,17,18]. However, in the absence of external forces, the high interfacial tension of the oil-water interface restricts this model to closely circular shapes. Taken together, there is still a lack of generally available water-immersed millimeter sized tissue mimetic that is compatible with the developments in the synthetic cell field and is able undergo dynamic remodeling events.

Here we introduce a lipid-based tissue mimetic that can be readily prepared and functionalized. The structural basis for our tissue mimetic is the assembly of a millimeter sized "lipid-foam" with individual micrometer-sized compartments. Biological tissues are often described as a type of active foam in physical models[19]. Throughout this work, we identify the connection between the foam-like properties of our mimetic tissues and biological tissues. We demonstrate that the tissue-mimetic has self-healing capabilities enabled by dynamic reorganization of subcompartments. We show that active tissue fluctuations through encapsulation of bacteria drive remodeling events and generate forces onto the environment.

This work is divided into three parts. First, we describe the assembly of a millimeter-scale lipid foam using microfluidics and phase transfer. Second, we characterize the assembly rules and mechanical properties of the resulting structure. Third, we induce active tension fluctuations by swimming bacteria, which we show drive large-scale fluidization and reorganization of the tissue mimetic.

## Results and discussion

Our initial goal was to create a suitable structural scaffold for the prototissue. We achieved this by adapting existing methods of flow-focusing microfluidic droplet formation[20], DIB[21–23], and emulsion phase transfer[24,25]. On a microfluidic chip, water-in-oil emulsion droplets were prepared and stabilized with the surfactant Span-80 (Fig. 1a). As expected for this established preparation procedure the generated droplets were monodisperse when generated at the microfluidic T-junction. Across experiments, droplet sizes varied between 20 and 50 μm depending on the flow rates of the fluids. A concentrated droplet emulsion was transferred into mineral oil that contained the phospholipid 18:1 (Δ9-Cis) PC (DOPC). Volumes were chosen such that DOPC was in similar concentration to Span-80 in the oil phase and equilibrated to form a self-assembled monolayer of DOPC molecules on the droplet surface. The mineral oil comprised the top layer of a two-phase liquid system with a water layer at the bottom (Fig. 1a). The resulting droplet concentration in the oil phase was chosen such that approximately a single layer of droplets sedimented at the oil-water interface. Droplets were then transferred into the water phase via centrifugation. The result was a structure consisting of millions of aqueous compartments, which were immersed in an outer water phase and overall spanned many millimeters in lateral size (Fig. 1b). The compartments formed were within the size range of the initial droplet preparation, but their size distribution was found to be more polydisperse. This suggests that while most droplets transferred to the water phase, a fraction underwent fission, fusion, or bursting during the assembly process. Nevertheless, the initial preparation of a homogeneous water-in-oil droplet was crucial for successful assembly. An initially heterogeneous emulsion (prepared by shaking) did not yield comparable structures after centrifugation (Supplementary Fig. 1). Additionally, the surfactant exchange established in this protocol was essential. The addition of lipids to the microfluidic chip caused droplet adhesion during downstream handling and pipetting, which interfered with the uniform packing of droplets at the oil-water

interface. Span-80 was necessary to obtain a stable water-in-oil emulsion, and the later addition of lipids provided the beneficial stabilization of droplets at the oil-water interface. When handled carefully during assembly, the resulting structures were relatively monodisperse in size and remained stable for days without significant structural degradation (Supplementary Fig. 2).

We next characterized the structural properties of the tissue mimetic. The prototissue is approximately two-dimensional—its lateral size is on the scale of millimeters and its height is in the range of 20–100 μm. However, the structure was not always a single monolayer —we often observed two or more layers of compartments stacked on top of each other. The optical density of the compartments did not allow us to study these stacked elements in detail, so we focused on the bottom layer, which appeared as a hexagonal lattice with relatively frequent packing defects (Fig. 1b). Fluorescent images show straight lamellae between individual compartments (arrow in Fig. 1b). The fluorescent dye used to visualize the compartments is a lipid analog. To confirm the lipids assembled into bilayers between compartments, we introduced the protein α-hemolysin (αHL) into the surrounding water phase. In lipid bilayers, but not thicker emulsion shells, αHL monomers assemble into membrane pores[26,27]. We monitored the permeability of the compartment boundaries using a membrane-impermeable dye. Only in the presence of αHL did the dye intensity decrease over time (Supplementary Fig. 3), confirming that the compartments are separated by lipid bilayers with a thickness of 3–5 nm. Another visible feature in the microscopic images is that bilayer segments intersect to form three-way junctions (Fig. 1b). We refer to these junctions as vertices. To characterize the nature of the contact between lipid bilayers and vertices, we performed fluorescent bleaching experiments (FRAP). Bleaching a rectangular section of compartments encapsulated with water-soluble dye (as shown in Fig. 1c), we observed that the bilayers were intact and impermeable to water-soluble molecules as expected. FRAP of the lipid analog dye resulted in the recovery of fluorescence within minutes, demonstrating that the lipid bilayers are fluid (Fig. 1d). Interestingly, only up to half of the original fluorescence was recovered, indicating that only one monolayer of the lipid bilayer had recovered its fluorescence. This incomplete recovery suggests that lipid transfer between compartments is not possible and that a single continuous monolayer encloses each compartment. While lipids are confied to the individual compartments, smaller, hydrophobic molecules (e.g., residual oil and surfactant Span-80 from the phase transfer procedure) might diffuse in the continuous bilayer/vertex phase (Supplementary Fig. 3 and Supplementary Note 1). From these experiments, we concluded that the microscopic configuration of a vertex consists of three lipid bilayers unzipping to monolayers with a Plateau–Gibbs border (Fig. 1e). This implies that lipid bilayers are under tension, consistent with no visible out-of-plane fluctuations of lipid bilayers, implying significant membrane tension that competes with thermal undulations (Fig. 1e). We quantified the bilayer tension experimentally using a water-filled micropipette (Fig. 1f). The micropipette was connected to an external pressure regulator, which allowed us to apply varying suction pressures $\Delta P = p_1 - p_0 < 0$. From Laplace law we calculated the corresponding surface tension $\gamma_c$ of the aspirated bilayer segment. At a critical tension, a compartment aspirated at edge of the structure was found to abruptly flow into the pipette, similar to a mechanical instability known for other soft matter systems under suction pressure (Fig. 1g)[28]. Inspired by these earlier works, we assumed that the measured critical tension $\gamma_c$ at the onset of flow is the value of $\gamma_B$[29,30]. To verify the validity of the measured tensions, we considered an established model of the bilayer tension generation by a force balance between the monolayer $\gamma_{ML}$ and the lipid bilayer $\gamma_B$ tensions[31]:

$$\gamma_B = 2\gamma_{ML} \cos(\phi) \qquad (1)$$

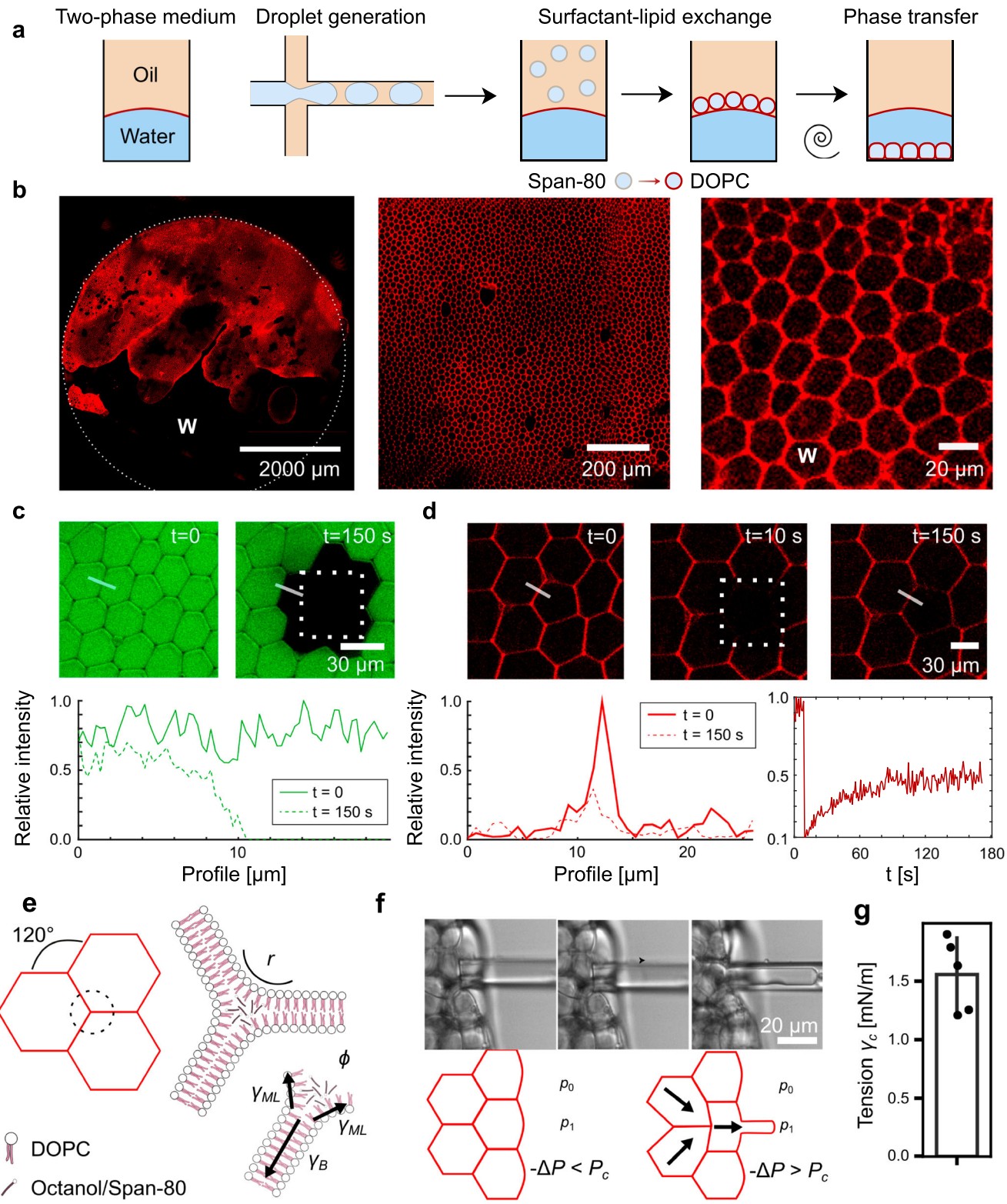

**Fig. 1 | Characterization of lipid-based prototissue. a** Schematic for preparing tissue mimetic structures. **b** Tissue mimetic structures consist of relatively homogeneous compartments that range in size between 20 and 50 μm in diameter and span many millimeters in lateral size. Examples at three different locations and magnifications are shown. Red indicates lipid analog fluorescence. Arrow highlights compartment boundaries as discussed in the main text. The letter W indicates the water phase. **c** FRAP experiment performed on compartments encapsulated with water-soluble dye (green). Fluorescence did not recover over the experimental timescale, indicating dye impermeability. Region bounded by dotted lines indicates photobleaching area. Plot shows fluorescence intensity profile along the path of the white line. Micrographs show representative images of successful preparations, see also section Statistics and Reproducibility. **d** FRAP experiment performed on lipid dye channel (red). Region bounded by dotted lines indicates photobleaching area. Fluorescence was partially recovered, indicating that only one monolayer of the lipid bilayer is connected to neighboring compartments (left). Fluorescence recovery over time is shown (right). **e** Schematic of lipid junctions between compartments, 1/*r* is the interface curvature and generally below optical resolution. **f** Time series (phase contrast images). i–iii show flow of a compartment into the micropipette at constant suction pressure (arrow indicates compartment edge). **g** Critical tension value at which compartment flow into the micropipette. Data are presented as mean values +/− SD. Each datapoint is an individual aspiration experiment from a total of *n* = 3 replicates. Source data for (**c, d, g**) are provided as Source Data file.

Typical values for lipid/surfactant monolayer to water interfacial tension $\gamma_{ML}$ are on the order of 1 mN/m. Together with the measured data for $\gamma_B$ (Fig. 1g), this implies a microscopic contact angle $\phi \approx 40°$. In black lipid membrane bilayers, that show the same monolayer and bilayer geometry, a comparable value of $\phi \approx 52°$ has been found[32,33]. We conclude that the in-plane bilayer tension is indeed in the range of a few mN/m and that the tension is generated by bilayer intermolecular adhesion energy between the leaflets. This intermolecular adhesion energy acts as a force that zips up the bilayer leaflets. It is important to note that the microscopic contact angle should be distinguished from the angle at which the bilayers meet at a vertex—specifically, the angle at which the lipid bilayers meet is always close to 120° consistent with a force balance between the three bilayers (Fig. 1e). Confluent biological tissues exhibit similar configurations, and rich frameworks exist to map biological tissues on traceable physical models. Specifically, comparison to the active foam model of tissues identify the bilayer tension $\gamma_B$ with the adhesion energy between cells and $\gamma_{ML}$ with contractile forces, that both exhibit similar magnitudes in tissues[19]. The active foam model predicts a confluent packing for values of $\gamma_B/\gamma_{ML} \geq 0.23$ consistent with our results. Connecting to earlier works that predict a density-independent rigidity transition in biological tissues, corresponding to $\frac{\gamma_B}{\gamma_{ML}} = 2$, we conclude that the here studied system is in a jammed state but positioned rather close to the transition, a feature we will explore further below[34]. It is also instructive to compare our system to foams in the dry limit, where the vertices studied here correspond to a collapsed Plateau border. Indeed, in most experiments, the curvature $1/r$ at the vertices could not be resolved, as expected from the dry limit where the curvature at the vertex is on the order of the lipid bilayer thickness. However, we occasionally observed larger oil pockets at the vertices, corresponding to Plateau borders that are "decorated" (Supplement Fig. 5). In this case, vertices contribute a stabilizing role; in other words, they have a negative line tension on the order of $\gamma_B r$[35]. Both comparisons, tissue models and foam, lead to the conclusion that the system is energetically stable, consistent with the experimental observations. This analysis highlights three crucial components of our tissue mimetic: membrane fluidity, stabilizing forces by the interplay of geometry and lipid self-assembly, and its biocompatibility.

In the aspiration experiments, we noticed the remarkable remodeling abilities that give the structure stability against perturbations. Exceeding the critical tensions, the aspirated compartment was observed to be replaced by neighboring compartments that flow into its former position (Supplementary Movie 1), triggering a series of remodeling events. This behavior has been also found in more high-throughput studies remodeling events in emulsion-based tissue models[18]. Tissue remodeling and fluidization is enabled by exchange of neighboring cells by so-called T1 transitions[8,36]. A T1 transition is a topological rearrangement event where one cell edge shrinks to a point and then expands in a perpendicular direction, resulting in the swap of neighboring cells. To trigger deformations larger in scale compared to the aspiration experiments, we mechanically loaded our structure by a glass capillary (Supplementary Movie 2 and Fig. 2a i). When we divided neighboring compartments by moving the glass capillary into the plane of the structure, we found that the compartments stayed intact. We observed that the structure reacted to the mechanical loading and within seconds to minutes the compartments reorganized to minimize their surface energy (Fig. 2a ii). When brought back into contact, the split structure resealed in a self-healing manner (Fig. 2a iii–iv). In this process, we observed that individual compartments could pass each other, resulting in compartments exchanging positions by a T1 transition (Fig. 2b). We interpret this series of events as loading above the yielding stress, which was sufficient to fluidize the otherwise jammed structure. The physical basis for the remodeling ability is the fluidity of lipid membranes and the ability of two monolayers to slide relative to each other in a process

called intermonolayer slip. Values for the intermonolayer slip range from $10^6$–$10^7$ dyn s cm$^{-3}$ [37] indicate the viscosity of the composite structure. These results showed that we can mimic the ability of biological tissues to react to external mechanical loading by inducing dynamic reorganization.

So far, we have characterized the lipid foam scaffold in its equilibrium properties and passive response to external mechanical loading. Yet, cells also exert forces on their environment, resulting in dynamic changes of tissue stresses, which ultimately drive tissue reorganization. In previous studies, membrane tension has been induced by collisions between active particles, such as chemically fueled swimmers or bacteria, and lipid bilayers[38,39]. Therefore, we hypothesized that by encapsulating bacteria into the compartments, we could induce active stresses in the tissue-like scaffold. We found that our encapsulation procedure successfully produces compartments with live, swimming *Bacillus subtilis* bacteria (Supplementary Movie 3). Visual inspection showed that the swimming bacteria collided with the bilayer membranes, transferring energy between the bacteria and membranes. We measured the swimming speeds of the bacteria to $3.5 \pm 0.5\,\mu m/s$ (std. dev., $n = 9$), by optical tracking. Additionally, the optical tracking indicated a persistence time of the swimming motion on the order of 1 s. This suggests that within the 20–50 μm diameter compartments, many collisions occur on the timescale of minutes. We reasoned that over longer timescales, the stresses induced by the bacteria might induce remodeling events. Consequently, we observed our hybrid synthetic-bacterial structure over many hours. Strikingly, the generated stresses seemed to drive T1 transitions from within the tissue (Fig. 2c and Supplementary Fig. 6). To understand the factors influencing the appearance of T1 transitions we investigated the generated stresses in more detail. Towards this goal we obtained detailed statistics of individual vertex positions. The rationale for tracking vertex position is that it is dependent on the force balance between the three bilayer tensions; if unequal tensions between the three bilayers were developed, the vertex would be displaced from its equilibrium position (Fig. 2c). With this assumption, the displacement of vertices from their average positions was chosen to be a measure of the prototissue tension. As expected for a finite temperature, empty compartments or heat-killed bacteria exhibited thermally driven length fluctuations. However, vertex fluctuations in samples of live swimming bacteria were significantly enhanced compared to samples of heat-killed bacteria (Fig. 2d and Supplementary Fig. 7). We further analyzed the power spectrum of both heat-killed and active bacteria and found that, within the sampled spectrum, the active systems exhibit higher amplitudes but overall similar shape of the spectrum (Fig. 2e). Thus, for our initial analysis we considered an effective temperature model of overdamped vertex fluctuations in a harmonic potential (Methods and Supplement Note 2). We found that the dynamics of the prototissue were relatively slow with a relaxation frequency $f_0 \sim 10^{-2}$ Hz, with a large drag coefficient $b \sim 0.5$ μNs/m reflective of the lipid bilayer viscosity. These values were comparable to relaxation timescales found in in-vivo tissue measurements in the order of $10^{-1} - 10^{-2}$ Hz[40]. The increase in effective temperature of $\frac{T_{eff}}{T} = 6.5$ by the encapsulated bacteria is comparable in magnitude to those found in biological systems. For example, ATP-driven membrane fluctuations in red blood cells result in $\frac{T_{eff}}{T} = 3$[41,42], underlining the fact that the high densities of bacteria encapsulated here provide a strong non-equilibrium driving force.

As two vertices are linked together by the lipid bilayer's edges, we further analyzed the change in distance between two neighboring vertices, the deviation from the average edge length (Fig. 3a). Again, significant broadening of the distribution was observed with swimming bacteria. The increase in length fluctuations corresponds to active tension dynamics studied in zebrafish embryos[19]. Both in biological tissues and the systems studied here, the largest length fluctuations are comparable to the length of an individual edge. These

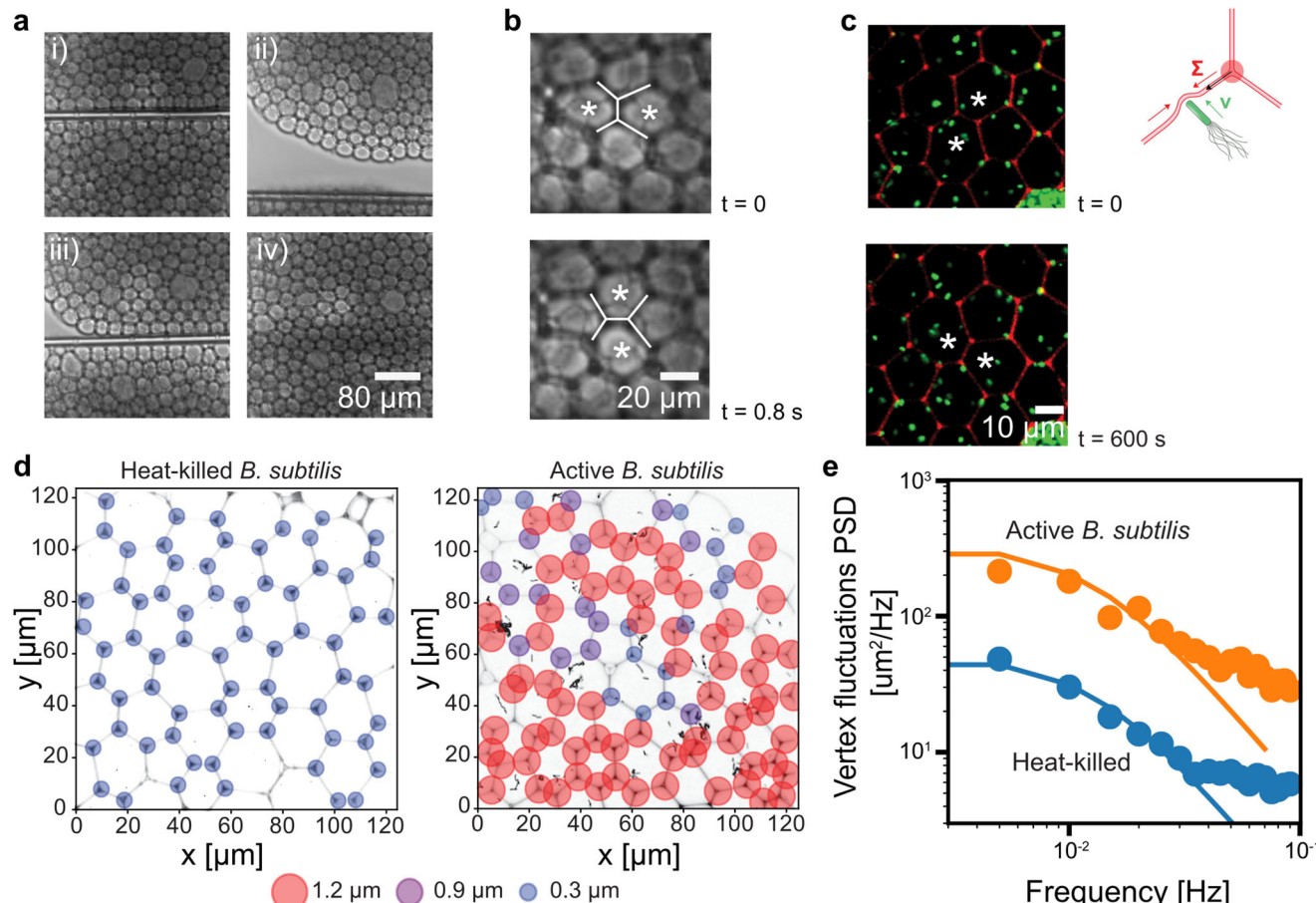

**Fig. 2 | Dynamic remodeling of prototissue enabled by lipid membrane fluidity.** **a** Series of events showing remodeling after mechanical loading: (i) scission by mechanical loading (ii) induction of a defect by movement of the pipette (iii) compartments are brought back into contact (iv) removal of pipette and healing of the structure. Micrographs are snapshots of Supplementary Movie 2 and is representative of $n = 3$ repeats of similar re-modelling events (**b**) Enlarged view of a T1 transition (shown in white) observed during step iv. **c** Bacteria (green) were encapsuled in compartments (lipid dye in red). ~~Spontaneous~~ T1 transition observed with encapsuled bacteria. Stars indicate two neighboring compartments before ($t = 0$ s) and after ($t = 600$ s) the remodeling event. Schematic shows how vertex displacement relates to membrane tension generation by collision of bacteria with lipid bilayer. **d** Spatial mapping of measured vertex displacement. Size and color of circles indicates the amplitude of the largest observed displacement over a time-frame of 10 min. **e** Vertex fluctuations power spectral density (PSD) from two $n = 2$ repeat experiments for each active and heat-killed condition. Straight lines are Eq. 1 in Supplementary Note 2 with parameters $b = 0.5$ μNs/m, $f_0 = 0.9 \times 10^{-3}$ Hz and $T_{eff} = 6.5$. Source data for (**d**, **e**) are provided as Source Data file.

large and rare displacements, spanning many microns, fall in the range of vertex distances and would be sufficient to drive T1 transitions[19]. However, between experimental replicates, we observed significant scatter in the distribution widths. We wanted to assess whether the initial preparation of the tissue structure during the centrifugation step might be contributing to this scatter. Therefore, we further analyzed the length fluctuations as a function of the shape factor of the polygonal compartments obtained from the microscopy images. The shape factor is defined by their perimeter $p$ and area $A$ via $s = \frac{p}{\sqrt{A}}$. We found the shape factor varied between experimental replicates because of the heterogeneity introduced by variation of individual compartments shapes (Supplementary Fig. 8). Overall, the studied shape index values ranged from $s = 3.77$, which is close to perfect hexagonal packing ($s = 3.722$), to larger values up to approximately $s = 3.8$. The largest values were close to the density-independent rigidity transition at $s = 3.81$[43]. And notably in the heat-killed samples, the distribution of length fluctuations appeared broader with increasing shape factor (Fig. 3b). This was consistent with a previous numerical study, which showed a decrease in the energetic barrier for T1 transitions target shape index[34]. For all shape factors, the swimming activity broadens the length distributions and thus

increases the chances of a T1 transition (Fig. 3b). Considering the shape factor did explain some of the variability between experimental repeats. However, still scatter between replicates was seen. This prompted us to further investigate the interactions between bacteria and compartments.

To understand the microscopic origin of stress generation by bacteria, we first considered a detailed model of active membrane mechanics in which membrane segments undergo active motion induced by the bacteria, leading to deformations of the membrane upon collision with bacteria in the normal direction of the bilayer plane. Parameterization of the model by the measured membrane tension $\gamma_B$, typical values for lipid bilayer bending rigidity, and bacteria velocity predicted only nanoscopic shape changes of the membrane by the swimming bacteria (Methods, Supplementary Fig. 9a). This is consistent with previous studies that find that only membranes with tensions below μN/m are significantly deformed by active swimmers[39]. These results indicated that we could directly map the swimming pressure exerted by the bacteria on compartment edges onto the vertex positions (Supplementary Methods, Supplementary Fig. 9b). We were initially surprised to find that in this model, equal loading of the three compartments with the same number of swimmers did not enhance vertex fluctuations on average (Fig. 3c). Instead, only an

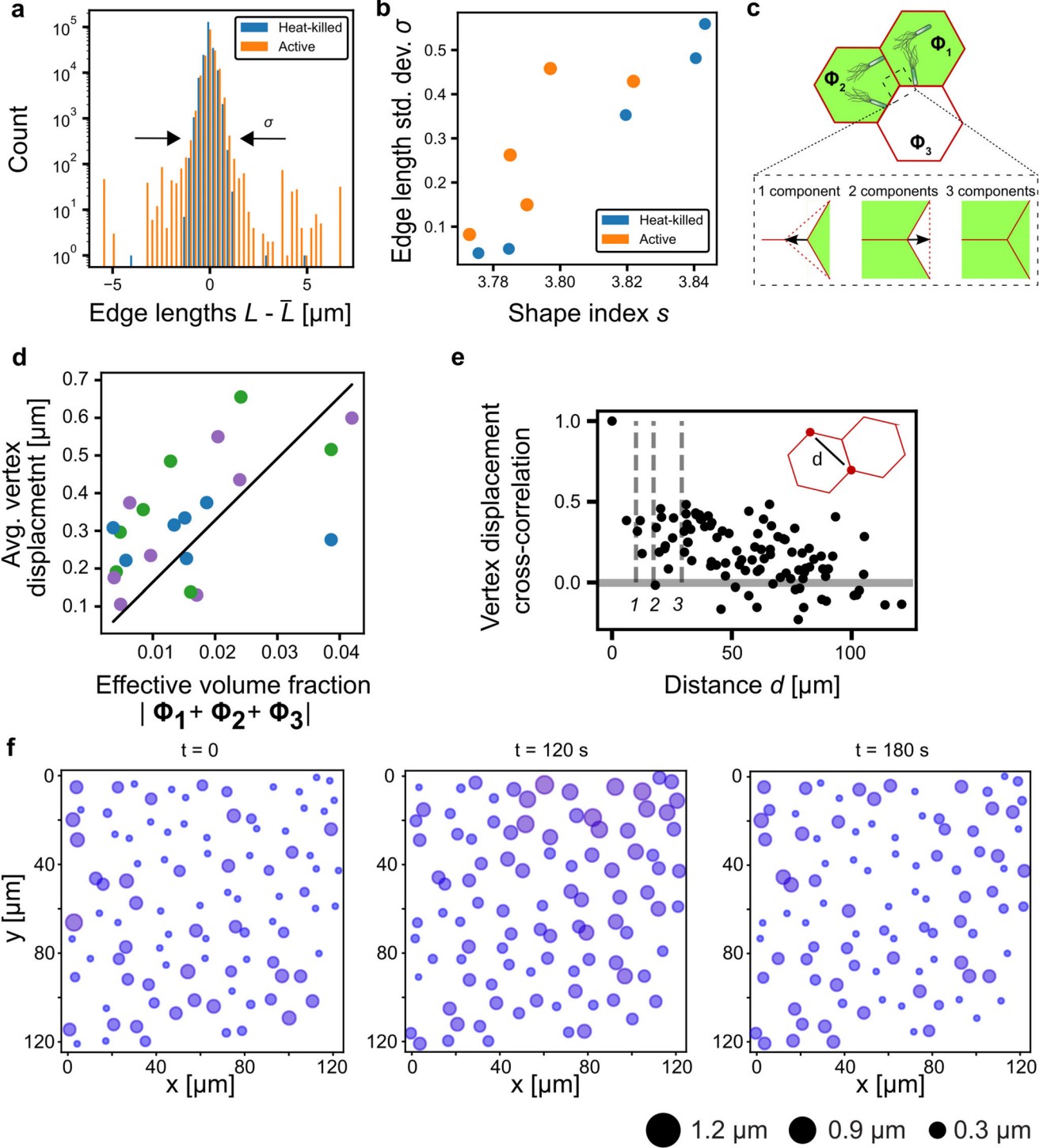

**Fig. 3 | Coupling of bacterial swimmers in an elastic lattice. a** Histogram showing the deviation of edge lengths $L$ from the mean length value $\bar{L}$ obtained for active (orange) and heat-killed bacteria (blue). Representative example of $n = 10$ experiments. Bacteria were encapsulated at an optical density of OD = 0.95, 600 nm. **b** Standard deviation of the $L - \bar{L}$ distribution for different values of shape index $s$ for active and heat-killed bacteria. Each datapoint represents an independent experiment ($n = 10$) **c** Examples of possible interactions between bacterial swimmers in an elastic lattice. A single compartment contains swimmers (green area), which leads to a vertex displacement (blue arrow) via the contraction of one edge and extension of two edges (dotted lines). When two compartments contain swimmers, the interaction leads to a vertex displacement as indicated by the blue arrow. When all three compartments contain swimmers, the interaction leads to zero displacement of the vertex. **d** Average individual vertex position displacement as function of vectorial volume fraction of bacteria in the three compartments sharing a vertex. Linear fit of slope 16.4 µm, zero intercept, with coefficient of determination $R^2 = 0.76$. Datapoints from a total of $n = 3$ experiments, colors show individual experimental repeats. **e** Cross-correlation between two vertices separated by distance $d$ over the displacement time series. Dashed lines show typical distance between two vertices (1,2,3 from left to right) **f** Time series of maximum vertex displacement within 60-s segments. Sizes and color of circles are proportional to displacement magnitude. Source data for (**a**, **b**, **d**–**f**) are provided as Source Data file.

unequal number of swimmers drove vertex displacement on time-scales longer than the typical bacteria-membrane interaction. Specifically, we found that when two equally loaded compartments compete against a single compartment, the resulting force balance dictates the upper and lower bounds of possible vertex displacements (Supplementary Methods). We tested this prediction by measuring the bacterial density in each compartment adjacent to a vertex and plotted the vectorial force balance at the vertex versus the average vertex displacement (Fig. 3d). Inspired by previous studies that considered tension generation by active particles[44,45], we found the average vertex displacement to be proportional to

$$\Delta\vartheta \sim \frac{3\eta\tau A}{4R^2}\frac{v^2\Phi}{\gamma_e} \qquad (2)$$

with a prefactor on the order of 1 that depends on the geometry of the deformation mode (Supplementary Methods). The values of swimmer velocity were measured $v = 3.5\,\mu m/s$, swimmer persistence time $\tau = 1\,s$, water viscosity $\eta = 10\,mPa$. We assumed a typical area of a bilayer segment of $A = 200\,\mu m^2$, the tension $\gamma_e = 3.1\,10^{-8}\,N/m$ was deduced from the measured passive length fluctuations $\gamma_e = \frac{k_BT}{\sigma}$ for the shape index in the range $s = 3.82$ (see Fig. 3b), which is close to the shape index for which data in Fig. 3d was collected. Note that this tension is much lower than the mechanical tension of the lipid bilayer $\gamma_b$. To map the bacterial occupancy $\Phi_n$ of each of the three compartments neighboring a vertex, as measured from the micrographs, we defined a vector $\Phi_n = \Phi_n(\cos\varphi_n, \sin\varphi_n)$, $n = (1, 2, 3)$, with $\varphi = (0°, 120°, 240°)$. The effective volume fraction occupied by the swimmers was then calculated by $\Phi = |\Phi_1 + \Phi_2 + \Phi_3|$. Here we exploit the stochastic variation of swimmers per compartment between 0 and 40 bacteria to sample a range of $\Phi$ values by a single experiment (Supplementary Fig. 10). Larger values of $\Phi$ indicate larger differences in bacterial occupancy and thus larger potential for vertex displacements. In our model the only unknown is the radius of the swimmer-lipid bilayer interaction $R$. Fitting the observed averaged edge displacements to the model we found $R \sim 2\,\mu m$, in the size range of a single bacteria, as expected (fitted line in Fig. 3d). This result also shows that to generate the largest possible tension (which effectively fluidizes the prototissue), control over the exact filling of the individual compartments with active swimmers is desirable. Such a method is currently not possible with the methods presented here. Taken together, we show that the magnitude of the generated tension is consistent with the swimming pressure generated by the bacteria and provide design rules that guide further development of the prototissue.

However, the model still could not explain all the scatter seen in experimental replicates, as evidenced by the deviations in Fig. 3d, which we estimate to be greater than the experimental accuracy of volume fraction determination and vertex displacement tracking. Even though our minimal model predicts a scatter in vertex displacements due to the asymmetry of extension vs. contraction events (Supplementary Fig. 9c), we reasoned that cooperative interactions across compartments could also play an additional role. Indeed, in vertex models with heterogeneous mixtures of cells such long-range correlations in the tension network have been found to dictate the tissue-scale mechanical features[46]. To systematically assess the effect of cooperative events, we measured the temporal cross-correlation between vertex displacements. A cross-correlation between the displacement of an individual vertex and every other vertex revealed a positive correlation in vertex motion up to a distance $l_{corr}$ of about $50\,\mu m$ (Fig. 3e). This suggests that bacteria from neighboring compartments indeed generate tensions cooperatively. As each compartment has on average six neighbors, cooperative events might generate significant "hotspots" of excitations as shown in Fig. 3f. Here, an initially relaxed configuration undergoes a transient state consisting of a collective motion of many vertices which then relaxes back over a timescale of a minute. Such a process highlights the role of the elastic lattice network that allows for cooperative interaction between the swimmers.

In this work, we have introduced a prototissue based on a biocompatible lipid foam scaffold. We balance the competing requirements of a tissue mimetic by preserving membrane fluidity and strong stabilizing forces that emerge from the interplay of geometry and lipid self-assembly. We have established a hybrid microfluidics-centrifugation protocol for the preparation, although it introduced some heterogeneity in compartments sizes and packing. Here, an all-microfluidic preparation might be desirable, but it currently seems unclear how to preserve the large-scale packing of the compartments and the commonly used junction chips that result in the formation of individual GUVs instead of the tissue mimetic obtained here. By analogy between foams and tissues, we have described its remodeling abilities. Our results demonstrate how the individual motions of bacteria can be used to induce coordinated motion in structures that are about two orders of magnitude larger than individual bacteria. In the future, better spatial control of the active swimmers and their distribution between compartments is desirable. Here, swimmers that are sensitive to optical or electrical fields provide ways to locally induce stresses that would allow to control and steer remodeling events.

## Methods

### Materials

1,2-dioleoyl-sn-glycero-3-phosphocholine (DOPC) and 1,2-dioleoyl-sn-glycero-3-phosphoethanolamine-N-(lissamine rhodamine B sulfonyl) (ammonium salt) (Liss Rhod PE) were purchased from Avanti Polar Lipids. D-glucose, D-sucrose, mineral oil, Span-80 surfactant and phosphate-buffered saline (PBS) tablets were obtained from Sigma-Aldrich. MSgg growth medium contains 5 mM potassium phosphate buffer (pH 7.0), 100 mM MOPS buffer (pH 7.0, adjusted using NaOH), 2 mM MgCl$_2$, 700 µM CaCl$_2$, 50 µM MnCl$_2$, 100 µM FeCl$_3$, 1 µM ZnCl$_2$, 2 µM thiamine HCl, 0.5% (v/v) glucose and 0.5% (w/v) monosodium glutamate, and were prepared using stock solutions at the current time.

### Bacteria strain generation

Custom promoter sequence (based on native *B. subtilis rpsD* promoter) was ordered from Integrated DNA Technologies (IDT) and cloned upstream of a YFP reporter gene sequence in *B. subtilis* integration vector ECE174. Plasmid assembly was conducted using Gibson Assembly with the Gibson Assembly Kit from New England Biolabs. Assembled plasmid was transformed into wild-type *B. subtilis 3610* using a natural competence protocol described by Ref. 47 and selected with appropriate antibiotic on LB agar plates.

### Lipid mixture preparation

Liss Rhod PE was added to DOPC in chloroform to make a 4.6 mM lipid mixture. The mixture was dried under a nitrogen stream to create a thin film on the bottom of a 20 mL scintillation vial. Lipid films were placed in a vacuum chamber for 2 h. The films were then resuspended in mineral oil and sonicated for 30 min in an ultrasound bath at 23 °C, resulting in a 19 mM DOPC in mineral oil mixture.

### Droplet generation using microfluidics

Within a well of a glass-bottomed 96-well plate, a two-phase oil-water mixture was created. This mixture consisted of 150 µL of DOPC in oil, which consisted the top layer, and 150 µL of aqueous glucose (600 mM), which consisted the bottom layer. Droplets averaging ~20 µm in diameter and containing aqueous sucrose (600 mM) in 4% Span-80 in mineral oil were generated on a PDMS T-junction microfluidic chip (Darwin Microfluidics). Using automated syringe pumps (Harvard Apparatus), the sucrose and 4% Span-80 in oil streams were flowed at 1 µL/min and 4 µL/min, respectively. Droplets were collected

in a pipette tip for 90 s before transferring to the oil phase of a loaded plate well.

### Creating droplets encapsulating bacteria

For bacterial encapsulation, a two-phase oil-water mixture was prepared similarly as described above, except the bottom water phase consists of 150 μL of PBS buffer (~ 300 mOsM). PrpsD10 *B. subtilis* NCIB3610 strains and MSgg media were prepared as described previously. The *B. subtilis* strain was grown in MSgg media for 16 h at 37 °C. Before encapsulation, the optical density ($OD_{600}$) was measured via UV–vis spectrophotometry (Agilent Cary 60), and the osmolarity was recorded with a vapor pressure osmometer (VAPRO) and diluted to the desired $OD_{600}$. To prevent osmolarity and density imbalances, D-glucose was added to the bacteria culture to match the osmolarity of the PBS solution. The osmolarity-adjusted bacteria suspension was then encapsulated into droplets as described above.

### Network formation via double emulsion

After droplet generation and transfer into wells, surfactant-lipid exchange was allowed to occur for 15 min. The droplets were then centrifuged at $3200 \times g$ for 5 min at room temperature, forming double emulsions of compartments at the bottom of the plate well.

### Imaging and image processing

Compartments were imaged using confocal microscopy (Nikon Eclipse Ti) using a 20× objective. Timelapses were captured at a frame rate of 1 frame per second and a total time of 10 min. Images of individual vertices were isolated, smoothed, and tracked in ImageJ (National Institutes of Health, Version 2.3.0). The vertex's position over time was tracked by identifying the pixel with the highest intensity in each frame (threshold tolerance = 50). To estimate bacterial occupancy $\Phi_n$ per compartment, confocal micrographs showing the fluorescent bacteria were binarized (ImageJ) and the fraction of (binary) pixels per compartment were counted.

### Graph reconstruction and downstream analysis

Using position information of the vertex and the tracking region-of-interest (ROI), membrane networks were virtually reconstructed using the igraph package (version 1.5.1) and tidyverse package (version 2.0.0) for *R* (version 4.2.3). For each timelapse, a time series of graphs, which contained position information and edge lengths, was created. After an initial reconstruction, the timeseries graph was visually simulated and inspected, removing from analysis any vertices showing signs of defective tracking.

Vertex fluctuation analysis was conducted using Jupyter Notebook and NumPy (version 1.20.1). Vertex tracking was confined to 1-min blocks to minimize the effects of drifting over long time scales. The deviation of the vertex position over time was taken to represent the amplitude of fluctuation. To create the fluctuation map, the position deviation of each individual vertex, across all time blocks, was taken. For data shown in Fig. 2d, to represent the characteristic amplitude of fluctuation for a single vertex, the maximum standard deviation was chosen. For data shown in Fig. 3d the average vertex distance from the center position was calculated. Edge fluctuation analysis was conducted using R. Edge fluctuation tracking was confined to 1-min blocks to minimize the effects of drifting over long time scales. The displacement of edge length ($L$) from the average length ($\bar{L}$) over the one-minute time block was taken to represent the amplitude of edge fluctuations. Using vertex position data and the information about the graph's connectivity, shape indices were calculated using the formula $s = A/\sqrt{P}$, where $A$ is the compartment area and $P$ is its perimeter.

### Statistics & reproducibility

Generally, an experimental replicate was considered as fresh preparation of lipid oil solution, microfluidic droplet preparation and

centrifugation. In about 15% of preparations no stable structures were obtained, likely due to problems in handling the emulsion or introduced impurities. When structures were unstable or did not form these experiments were excluded from further analysis. No statistical method was used to predetermine sample size. The experiments were not randomized. The Investigators were not blinded to allocation during experiments and outcome assessment.

### Theoretical description of membrane deformations driven by bacteria

We considered a model of active particles on a membrane for evolution of the membrane height function[48–50], and solved the equations numerically using the Python package py-pde (version 0.42.1)[51]. Derivations and further details are found in the Supplementary Methods.

### Reporting summary

Further information on research design is available in the Nature Portfolio Reporting Summary linked to this article.

## Data availability

Data supporting the findings of this study are available within the paper and its Supplementary Information files. Source data for all figures (in the Main Text and Supplementary Information) are provided as a Source Data file. Source data are provided with this paper.

## Code availability

Analysis code in *R* and python is available as a GitHub repository[52].

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

## Acknowledgements

This research was supported in part by the National Science Foundation under Grant No. 1844336 (J.S.), 2239567 (A.P), and MRSEC DMR-2308691 (A.G., N.P.K.) and the National Institutes of Health under Grant No. 1R35GM147170-01 (A.P). J.S. thanks Reinhard Lipowsky for discussions on stability of foams.

## Author contributions

A.G. carried out tissue mimetic synthesis, image analysis, and graph reconstruction experiments. J.S. performed initial experiments, structure determination, micropipette experiments and modeling. J.S. and N.P.K. conceptualized the experiments and analysis. PrpsD10 *B. subtilis* strains and optimization of encapsulation protocol were developed and provided by P.T. and A.P. Theoretical model for tension generation was developed by M.C.U. A.G., J.S., and M.C.U. jointly prepared the manuscript draft. All authors discussed and interpreted data and contributed to writing of the manuscript.

## Funding

## Competing interests

The authors declare no competing interests.
