## [Transparent Peer Review file · Nature Communications]

Remodeling of lipid-foam prototissues by network-wide tension fluctuations induced by active particles

Corresponding Author: Professor Jan Steinkühler

Version 0:

Reviewer comments:

Reviewer #1

(Remarks to the Author)

Gu and colleagues describe the construction of "proto-tissues", a very straightforward and interesting application with microfluidically mass-produced synthetic cells. The idea has recently been discussed quite frequently in the bottom-up synthetic biology field, but so far lacks a convincing realization that goes beyond proof-of-principle. Gu et al. propose so-called "lipid foams" as tissue mimicry, assembled from high numbers of giant lipid vesicles, and characterize these foams with regard to their structural and mechanical properties and the stability in terms of self-healing capabilities. The further include actively swimming bacteria in the foams and observe an interesting mechanic response of their proto-tissues. Overall, this is a nice assay, and there are many interesting aspects of the study that deserve further attention, but I do not consider this story a mature one at this point. I will raise my detailed concerns below.

1. The assembly of lipid foams:

- a) In caption Figure 1. (b) the statement "20 um in diameter compartments" requires a histogram to demonstrate this, or the statement should be changed. If the authors produce heterogenous populations of 20-50 um water-in-oil droplets, as indicated later, why not attempt their technique without the microfluidics generation of droplets for comparison?
- b) The authors claim the majority of the span-80 detergent is exchanged with DOPC. It is important for the authors to clarify the level of detergent remaining in their assembly and the importance of span-80 presence or absence in the assembly process.
- c) In the Results section, the statement "The structures assembled in the established conditions were remarkably stable and lasted days without obvious degradation in overall morphology" requires data to back the claim and whether the tissue changes its morphology in terms of reshaping autonomously over time.
- d) Fig. S1 in the SI raises more questions than it answers. Why would the hydrophilic dye used to study leakage from the compartments be sitting in the hydrophobic interface? What are the leakage time scales? Did the aHL remodel the foam? The control would have to be repeated at a much better quality.
- e) In the Results section the statements "hexagonal packed honeycomb lattice" and "The angle at which the lipid bilayers meet is always close to 120° consistent with a force balance equal compartment volumes, due to their microfluidic preparation, the bilayer tensions lead to the formation of an ordered lattice analogous to Plateau's laws in foams." are not very accurate in the case of what the authors are presenting, since many of the images displayed in the paper contain assemblies with (squares, pentagons and heptagons) with different volumes as pointed out earlier in a).

2. The remodeling lipid-foam induced by swimming bacteria:

- a) The phenomenon of using bacteria to remodel synthetic tissue documents only one incident of compartment remodeling between a pentagon, hexagon and heptagon (fig 2B), while (fig 2D and Mov2) demonstrates minimal/negligible displacement of the vertices that can fall under a sample to sample variance and would require repeats. The idea and story of remodeling synthetic tissue using active particles is interesting, but the execution and results are underwhelming and require great amount of effort to convince the general audience.
- b) Same goes to the self-cleaning capabilities of the tissue demonstrated in Mov3 and fig 3E, the majority of the beads are stuck to the foam for the effect to be taken seriously.

3. The manuscript would benefit from significant language and style revisions. The text is riddled with spelling and grammar mistakes which does not inspire confidence while reading.

Reviewer #2

(Remarks to the Author)

I co-reviewed this manuscript with one of the reviewers who provided the listed reports as part of the Nature Communications initiative to facilitate training in peer review and appropriate recognition for co-reviewers.

Reviewer #3

(Remarks to the Author)

I here report on the manuscript submitted to Nature Communications entitled 'Remodeling of lipid-foam prototissues by network-wide tension fluctuations induced by active particles' by Gu et al.

The manuscript deals with the formation of a model tissue and the study of its dynamics in presence of active particles (here living bacteria). The merit of the work is to include in the system an active component in the form of a bacterial suspension which modifies the structure of the tissue. In general the problem is of interest: understanding how activity controls morphologies and structures in soft matter systems is of relevance to better understand living matter in general and the role of the out of equilibrium conditions on (bio-)materials in particular.

The work is based on the generation of a prototissue using GUV produced in a hybrid process involving microfluidics and bulk phase transfer. The realization of artificial tissues of this kind have been produced by full microfluidic methods which are not mentioned (see eg doi.org/10.3389/fphy.2022.1014428 | doi.org/10.3389/fphy.2021.744006) and one might argue that a full microfluidics method would help to obtain even better controlled tissues.

The activity of the bacteria affects the network of vesicles which is measured experimentally by image processing. But the statistics of the rearrangement in the model prototissue - especially considering the T1 event - could be pushed a bit further. At present the work solely focus on vertex displacement, showing a shift of the fluctuations towards higher values in presence of bacteria. The T1 events are related to large displacement of the vertex but one might argue that a more direct determination of the T1 event and the statistics of their occurrence could be performed (see eg [10.1039/D1SM00097G](https://doi.org/10.1039/D1SM00097G)): single events, cascades and avalanches...

The underlying physics of reorganization remain at the qualitative level with no direct measurements of fundamental parameters related to the dynamics of the bacteria. For example, the velocities of the bacteria under this confinement are not directly determined. The stress induced by the motion of the bacteria in the droplet is also not measured. It has been reported that active particles in confined system modulate interfacial properties ([10.1103/PhysRevLett.129.138001](https://doi.org/10.1103/PhysRevLett.129.138001)). A quantification of the impact of the bacterial swimming on the GUV properties used in the paper would be useful. For example, we would need to understand how the swimming leads to the generation of stresses at the interface which the paper does not quantitatively discuss.

Experimentally, only two limiting cases are presented (with live and dead bacteria). Obviously a large number of bacteria are required to obtain a measurable effect. What happens when the activity is varied by changing simply the number of bacteria ? Is there a collective dynamics in the bacterial swimming required to induce the reorganization ? Without a more detailed analysis of the link between activity and reorganization, the reader is left with qualitative interpretations that are not fully satisfactory.

To conclude, at present, the experimental results presented here appear to be promising preliminary results that however do not seem to be sufficient to understand the (bio-)physics of the material rearrangement.

Version 1:

Reviewer comments:

Reviewer #4

(Remarks to the Author)

The authors have significantly expanded their manuscript in response to the comments from the reviewers. They have done additional experiments and complemented their data with a model. Overall, the authors have addressed the concerns well and the manuscript has clearly improved and now fits within the impact and quality criteria of Nature Communications. With the edits, some minor points for improvements arose and I detail them below.

1. The authors now reference the work of Pontani as references 8 and 11, but just mention it without providing its importance in the context in the manuscript, while pointing out in the reply that this is relevant for their work.
2. The new panel g in Figure 1 is showing 5 data points, of which four are connected by a line and one is off the line. Why does the line, which I assume from the caption to be the standard deviation, precisely connects four points? Why is one point next to the others – has it been measured differently, is it shown there for clarity?
3. Figure 3b misses a legend.
4. Figure 3e: It is unclear what the dashed lines refer to – “1,2,3” is not explained in the inset, for example.

5. I would suggest clarifying in the title that bacteria are used as "active particles".

Reviewer #5

(Remarks to the Author)

I have been asked by the editor as a replacement reviewer to review the revised manuscript by Steinkühler and coworkers, and to assess the response to the points raised by the original reviewers of this manuscript. I have limited myself to commenting on the responses and changes made to the manuscript, besides a few technicalities listed at the end. In general, the authors responded to the points raised by the reviewers in great detail and have significantly rewritten their manuscript. They performed additional experiments, added a newly developed theory, and removed a part that could not be repeated with sufficient fidelity. However, a few minor points remained unclear, as indicated below. I would recommend to revise the manuscript based on these points to further improve clarity.

Reviewer 1

1a. This response convincingly addresses the question about compartment size distributions and the importance of a homogeneous size distribution. As a note: the reviewer was confused about the statement "20-50 um diameter compartments", which suggests a heterogeneous sample. The authors may wish to clarify that they study samples with (relatively) homogeneous compartments, that range in size between 20 and 50 um. Within this size range, the collision frequency is of the order of 0.1/s.

b. The authors provide a reasonable estimate of the Span-80 concentration. However, it should be noted that their estimate does not take into account any oil pockets that remain in the final structures. It would be advisable to add a statement about the presence of oil pockets and, ideally, to estimate their contribution to the Span-80 concentration.

c. The new data supports the statement on stability.

d. While the bleed-through is not ideal (in hindsight, another dye would have been better suited), it indeed does not interfere with the interpretation of the images and the aHL experiment, combined with the bleaching and MPA data support the conclusion that the foam edges are lipid bilayers.

e. The introduction of the shape factor and the updated discussion are helpful to take away any confusion about the deviations from hexagonal packing and implications thereof.

2a. The authors have added more examples of T1 transitions and developed a theory that quantitatively describes vertex fluctuations. Although T1 transitions remain rare with swimming bacteria, the vertex fluctuations are significantly enhanced with swimming bacteria, and can be quantitatively described with a model that includes an effective temperature to account for the activity. These additions convincingly show that the tissue mimetic exhibits to network-wide tension fluctuations driven by membrane tension generation by the swimming bacteria.

b. This seems to be the correct thing to do.

3. This reviewer found the manuscript to be generally well written. A few minor points are listed below.

Reviewer 3

1. This addresses the point raised by the reviewer.

2. See also the point raised by Reviewer 1, sub 2a above. T1 transitions remain rare, and the authors chose to focus on vertex displacement, for which they have now included a theoretical description that quantitatively describes their data. This is a new aspect of the current study and it is now convincingly demonstrated.

3. The authors quantify some of the properties mentioned by the reviewer, and present a theory to achieve deeper insight into the impact of bacterial swimming on vertex fluctuations and reorganization. This point has been partly addressed. In particular, the authors should add a histogram to support their estimate of 10-100 bacteria per compartment (ϕ). In addition, the newly added section in which the influence of the bacterial occupancy is discussed (page 9 and Fig. 3c) is not entirely clear. Fig. 3c appears to be based on experimental data from three samples, but it is not clear how the volume fraction was estimated, how it was varied between 0.01 and 0.04 (is this a poisson distribution), which data points correspond to which experiment. Moreover, the text suggests this figure shows data for equal loading of the compartments ("We were initially surprised to find that in this model, equal loading of the three compartments with an equal number of swimmers did not enhance vertex fluctuations on average (Fig. 3c)."). However, the figure does not show the average vertex fluctuations, but the standard deviation. Moreover, in their response to point 4, the authors write that "the differential filling of neighboring compartments is the dominant effect on vertex fluctuations." Does Fig. 3c show data for equal filling or differential filling, and if so, how is the effective volume fraction defined? The authors should revise the text and figure presentation and caption to

clarify these points.

4. See point 3. The theory can be used to extend the analysis to a wider range of cases, but some parts of the new discussion should be clarified.

Additional technical points and writing style:

1. Mix up of Span-80, span-80 and SPAN 80.
2. vertexes/vertices both used
3. Inconsistent references to Supplementary (Supplement Fig. 4, Supplementary Fig. S3, Supplementary Fig. 2).
4. Movie 3 does not appear to show the T1 transition, but the text somehow suggests it does: it is referenced together with Fig. 2c, which depicts a T1 transition. It would be advisable to include a video from which the snapshots in Fig. 2c were taken.
5. Typo: "repats" (caption of Fig. 2). Additionally, are these actually repeats? One curve seems to correspond to active and one to a passive system. The caption for Fig. 2e lacks information about the difference between the fits of the two series, and a reference to eq.2 for the fit is incorrect, this should be eq. 1 in Supplementary Note 2.
6. Typo: swimming speeds the bacteria ("of" missing)
7. Fig. 3a: frequency should have a unit.
8. Fig. 3b: y-axis lacks units. The meaning of the different symbols is not defined. Are these prototissues with and without bacteria?
9. Fig. 3f: a legend to correlate circle size with the actual displacement magnitude (number) is missing.

We thank all three reviewers for their time, critical remarks, and the opportunity to present a significantly extended and revised manuscript. To address the points raised, we have conducted new experiments, additional repeats, new analyses and extended the theoretical model. In total, we have added four new figure panels in the main text, seven new figures to the supplementary information, and four pages of new theoretical analysis. We believe that these changes fully address the reviewers' comments, and we detail the modifications made point-by-point in the following. Our responses are written in blue. Changes to the main text and Supplementary Information (SI) are highlighted in red.

Reviewer #1 (Remarks to the Author):

1. The assembly of lipid foams:

a) In caption Figure 1. (b) the statement “20 um in diameter compartments” requires a histogram to demonstrate this, or the statement should be changed. If the authors produce heterogenous populations of 20-50 um water-in-oil droplets, as indicated later, why not attempt their technique without the microfluidics generation of droplets for comparison?

We have added the new Supplementary Figure 1d, which shows a histogram of compartments sizes. We have amended the main text on page 4 to clarify that within a single preparation, the generated water-in-oil droplet sizes were homogeneous and that heterogeneity was introduced during handling. When handled carefully, the produced tissues exhibited quite monodisperse sizes, as seen in Supplementary Figure 1d. We have also added (on page 4) the results of our attempts to prepare the structure from a heterogeneous emulsion prepared by shaking. These attempts did not result in structure formation, likely due to the more disordered packing at the water-oil interface before the centrifugation step. This highlights the importance of the preparation protocol developed in this work. Finally, we have explored the effects of variability in compartment packing on vertex fluctuations (new Fig. 3b).

b) The authors claim the majority of the span-80 detergent is exchanged with DOPC. It is important for the authors to clarify the level of detergent remaining in their assembly and the importance of span-80 presence or absence in the assembly process.

We agree with the reviewer that a more detailed analysis was necessary. We have highlighted the role of Span-80 on page 4 and estimate that the concentration of Span-80 in the prepared structures is small (new Supplementary Note 1), but it cannot be fully excluded (page 5). This is consistent with the more refined analysis and experiments that measure the lipid bilayer tension (Fig 1g), which indicate that the lipid bilayers fully “zip up”—a process typically assumed to expel much of the residual solvent and surfactant. We have conducted initial tests using other oils and surfactants (1-octanol, oleic acid, squalene, Poloxamer 188 Non-ionic Surfactant) but have not managed to obtain similar structures (this data was not included in the manuscript), highlighting the importance of the detailed protocol established in this work.

c) In the Results section, the statement “The structures assembled in the established conditions were remarkably stable and lasted days without obvious degradation in overall morphology” requires data to back the claim and whether the tissue changes its morphology in terms of reshaping autonomously over time.

We have imaged a single prototissue structure over 44 hours and found only minimal changes in morphology, as presented in Supplementary Figures 1a to 1c. In the new Fig. 3b, we present quantitative measurements of vertex fluctuations for both active and passive structures, providing more quantitative differences between autonomous and induced remodeling.

d) Fig. S1 in the SI raises more questions than it answers. Why would the hydrophilic dye used to study leakage from the compartments be sitting in the hydrophobic interface? What are the leakage time scales? Did the aHL remodel the foam? The control would have to be repeated at a much better quality.

We thank the referee for pointing out this source of confusion. In the experiments presented in Supplementary Figure 2, the final dye concentration was relatively low, which required high laser intensity during imaging. This resulted in the bleed-through of the red-shifted fluorescent membrane dye in these experiments. Please note that the same water-soluble dye is shown in Fig. 1c, where the membrane dye was absent. These images show that the hydrophilic dye does not preferentially localize at the hydrophobic interfaces. In the presented aHL experiments, the bleed-through does not interfere with the interpretation of the images, which clearly show that aHL induces leakage of the hydrophilic dye from the aqueous compartment. The timescales and improved control images are now included in the revised Supplementary Figure 2. The possibility of aHL remodeling the foam structure is indeed very interesting, and some of the images seem to indicate this effect, as the bilayers of the permeated compartments appear to develop some curvature. The curvature should correspond to pressure differences between compartments. A possible mechanism could involve leakage of osmotic agents, but we have not studied these effects in detail. This is because the asymmetric insertion and assembly of aHL, leakage of osmotic agents, and possible remodeling of the tissue occur on similar timescales, making them challenging to study with the available methods. The manuscript's main focus is on the remodeling ability of individual compartments by active swimmers, and we have focused on this deformation mode.

We believe that the presented aHL experiments fully support the conclusion made: the structure of individual “edges” of the foam are lipid bilayers. This conclusion is also consistent with the bleaching experiment (Fig. 1d) and new quantitative micropipette aspiration data and analysis (Fig 1g, page 5).

e) In the Results section the statements “hexagonal packed honeycomb lattice” and “The angle at which the lipid bilayers meet is always close to 120° consistent with a force balance equal compartment volumes, due to their microfluidic preparation, the bilayer tensions lead to the formation of an ordered lattice analogous to Plateau’s laws in foams.” are not very accurate in the case of what the authors are presenting, since many of the images displayed in the paper contain assemblies with (squares, pentagons and heptagons) with different volumes as pointed out earlier in a).

Following the reviewers' comments, we have carefully reworded these statements and provided a significantly extended analysis. Specifically, we show a closely hexagonal structure in Supplementary Figure 1 and provide multiple repeated experiments where we use the "shape factor" to measure the deviation from completely hexagonal packing. These new experiments are presented in Fig. 2b. We have also added additional data to clarify the role of tension in the new Fig. 1g. Additionally, we have completely revised the discussion (page 5 and 6), and together with the updated sketch in Fig. 1e, this should now accurately present the results of the characterization.

The referee also pointed out that the transient 120° force balance is disrupted by the application of membrane tension, as observed in swimming bacteria. We have quantified these effects with a more complete theory of elastic deformation (new Supplementary Fig. 9C); please also see the response to reviewer 3 below.

2. The remodeling lipid-foam induced by swimming bacteria:

a) The phenomenon of using bacteria to remodel synthetic tissue documents only one incident of compartment remodeling between a pentagon, hexagon and heptagon (fig 2B), while (fig 2D and Mov2) demonstrates minimal/negligible displacement of the vertices that can fall under a sample to sample variance and would require repeats. The idea and story of remodeling synthetic tissue using active particles is interesting, but the execution and results are underwhelming and require great amount of effort to convince the general audience.

We have now significantly extended the number of repeats to present the conclusions more convincingly. In particular, we show three additional independent repeats in Supplementary Figure 5. Additionally, we have extended the analysis vertex fluctuations and present data from an additional six repeat experiments, leading to a total of $n = 10$ (active and passive) repeats in the new Fig. 3b. Together with the completely newly developed theory that details the interaction between active particles and the membrane (Supplementary Methods and Fig. 8), this quantitatively matches the observed displacements (new Fig. 3d) with a single fitting parameter.

b) Same goes to the self-cleaning capabilities of the tissue demonstrated in Mov3 and fig 3E, the majority of the beads are stuck to the foam for the effect to be taken seriously.

Given the additional data clearly showing enhanced fluctuations of the structure, it is evident that a bead in contact with the membrane would also experience some enhancement in movement. However, it is true that the self-cleaning capability was not demonstrated to the highest standard. We found it challenging to repeat these experiments with sufficient fidelity to provide more convincing results. Likely, tuning the lipid composition (charge, PEGylation, etc.) would be beneficial in reducing the pinning of beads to the structure. Considering the focus of the work (preparation, characterization, and demonstration of active remodeling of a new prototissue), we decided not to further pursue these experiments in the present study. We acknowledge that this remains, in our opinion, the only point raised by the reviewers that we could not fully address and have removed the data in former Fig 3e.

3. The manuscript would benefit from significant language and style revisions. The text is riddled with spelling and grammar mistakes which does not inspire confidence while reading.

We have carefully revised the whole manuscript and SI for spelling and grammar mistakes.

Reviewer #2 (Remarks to the Author):

I co-reviewed this manuscript with one of the reviewers who provided the listed reports as part of the Nature Communications initiative to facilitate training in peer review and appropriate recognition for co-reviewers.

We thank the referee for providing a co-reviewer with the opportunity to participate in peer review and for co-reviewing the manuscript.

Reviewer #3 (Remarks to the Author):

The work is based on the generation of a prototissue using GUV produced in a hybrid process involving microfluidics and bulk phase transfer. The realization of artificial tissues of this kind have been produced by full microfluidic methods which are not mentioned (see eg doi.org/10.3389/fphy.2022.1014428 | doi.org/10.3389/fphy.2021.744006) and one might argue that a full microfluidics method would help to obtain even better controlled tissues.

We thank the reviewer for pointing us to two interesting references that we were not aware of before. We have included these as new references 8 and 11 on pages 1 and 2. In particular, new reference 8 nicely highlights the importance of balancing adhesive forces and fluidity to obtain biologically realistic assemblies, which was indeed the motivation for the present study. We also agree with the reviewer that a full microfluidic method might be desirable, and we discuss the current limitations of microfluidic devices that might be used for prototissue assembly on page 11.

The activity of the bacteria affects the network of vesicles which is measured experimentally by image processing. But the statistics of the rearrangement in the model prototissue - especially considering the T1 event - could be pushed a bit further. At present the work solely focus on vertex displacement, showing a shift of the fluctuations towards higher values in presence of bacteria. The T1 events are related to large displacement of the vertex but one might argue that a

more direct determination of the T1 event and the statistics of their occurrence could be performed (see eg 10.1039/D1SM00097G): single events, cascades and avalanches...

Initially, we studied vertex displacement, as the length fluctuations of edges between two vertices need to be large enough to allow for a T1 transition in our system. The reviewer is correct that the mechanics of compartment movements should have been studied in greater detail. Therefore, we have included new experimental data in Fig. 1g that measures the lipid bilayer tension by micropipette aspiration and, at high tensions, induces remodeling events. Together with additional new analyses, this was instrumental in understanding the connection between the magnitude of vertex fluctuations, T1 transitions, and the packing of the compartments. Please also see the reply to Reviewer 1 below. We have included the cited work as new reference 35 on page 6, as it nicely demonstrates how the movement of individual compartments might trigger larger-scale events with universal dynamics. Indeed, similar events have been observed in this study (new Supplementary Movie 1). Given the very detailed characterization of scaling laws in reference 35, we have not attempted to obtain similar data for our compartments and instead focused on the new aspects of this study.

The underlying physics of reorganization remain at the qualitative level with no direct measurements of fundamental parameters related to the dynamics of the bacteria. For example, the velocities of the bacteria under this confinement are not directly determined. The stress induced by the motion of the bacteria in the droplet is also not measured. It has been reported that active particles in confined system modulate interfacial properties (10.1103/PhysRevLett.129.138001). A quantification of the impact of the bacterial swimming on the GUV properties used in the paper would be useful. For example, we would need to understand how the swimming leads to the generation of stresses at the interface which the paper does not quantitatively discuss.

We have significantly extended the manuscript to achieve a more comprehensive and quantitative study. To accomplish this, we have included a new author, Mehmet Can Uçar from IST Austria, who contributed significant theoretical insights into the physics of the prototissues studied here. These insights were used to parameterize the model presented in the Supplementary Methods and in the main text on pages 8 and 9. We have also incorporated the active swimmer model used in new reference 44, as suggested by the reviewer.

Experimentally, only two limiting cases are presented (with live and dead bacteria). Obviously a large number of bacteria are required to obtain a measurable effect. What happens when the activity is varied by changing simply the number of bacteria ? Is there a collective dynamics in the bacterial swimming required to induce the reorganization ? Without a more detailed analysis of the link between activity and reorganization, the reader is left with qualitative interpretations that are not fully satisfactory.

We now provide a quantitative analysis that fits the data with a single fitting parameter (new Fig. 3d, page 9, Supplementary Figure 8, and Supplementary Methods). Based on this model, we can conclude that the differential filling of neighboring compartments is the dominant effect on vertex fluctuations. By analyzing and mapping this effect on the new parameters of the effective volume fraction that acts on vertex displacement, we are able to extend the analysis to a wide range of cases.

We thank the new referees for their careful consideration of the previous reports and our revised manuscript. The reports have acknowledged our significantly extended manuscript, including a detailed theory of the swimmer-induced deformation and favorable comparison to experiments which have in principle addressed all requests. However, the new referees requested some additional clarifications and explanations, which we have fully addressed in detail. The point-by-point response follows.

Reviewer #4 (Remarks to the Author):

1. The authors now reference the work of Pontani as references 8 and 11, but just mention it without providing its importance in the context in the manuscript, while pointing out in the reply that this is relevant for their work.

We would like to fully acknowledge the important previous work of the Pontani group. To highlight the relevance of the previous work, we have added an additional discussion of their oil-in-water droplet system on page 2 of the revised manuscript. The work by Pontani et al is also referenced in relation to our work on page 6. In the new numbering, the three cited papers from the Pontani Lab are listed as refs. 8, 17, and 18.

2. The new panel g in Figure 1 is showing 5 data points, of which four are connected by a line and one is off the line. Why does the line, which I assume from the caption to be the standard deviation, precisely connects four points? Why is one point next to the others – has it been measured differently, is it shown there for clarity?

We thank the reviewer for identifying this potentially confusing presentation. The point in question was jittered by the plotting software to avoid overlap with the adjacent datapoint. This datapoint was not measured differently. We have now jittered all points away from the line indicating the standard deviation. Additionally, we have replaced the plot in Fig. 1g with a higher-resolution version. The presented data is unchanged.

3. Figure 3b misses a legend.

We have added a legend to Fig. 3b.

4. Figure 3e: It is unclear what the dashed lines refer to – “1,2,3” is not explained in the inset, for example.

We have added numbers to the dashed lines.

5. I would suggest clarifying in the title that bacteria are used as "active particles".

Given the already long title, we would prefer to keep the current title. In the abstract, we explicitly specify the use of bacteria as active particles. Additionally, bacteria are often referred to as active colloids/particles in the literature (see <https://doi.org/10.1016/j.colsurfb.2015.07.048>)

Reviewer #5 (Remarks to the Author):

(former) Reviewer 1

1a. This response convincingly addresses the question about compartment size distributions and the importance of a homogeneous size distribution. As a note: the reviewer was confused about the statement “20-50 μm diameter compartments”, which suggests a heterogeneous sample. The authors may wish to clarify that they study samples with (relatively) homogeneous compartments, that range in size between 20 and 50 μm . Within this size range, the collision frequency is of the order of 0.1/s.

We thank the reviewer for this additional opportunity for clarification, apparat from the already made changes in the main text we have added the information “Tissue mimetic structures consist of relatively homogeneous compartments in the range of 20-50 μm in diameter and span many millimeters in lateral size” to the figure 1 caption.

b. The authors provide a reasonable estimate of the Span-80 concentration. However, it should be noted that their estimate does not take into account any oil pockets that remain in the final structures. It would be advisable to add a statement about the presence of oil pockets and, ideally, to estimate their contribution to the Span-80 concentration.

We thank the reviewer for acknowledging our efforts to estimate the Span-80 concentration. While we are unable to provide an estimate for individual oil pockets, we have included a statement about the relevance of such pockets in Supplementary Note 1 on page 11 of the SI.

Reviewer 3

3. The authors quantify some of the properties mentioned by the reviewer, and present a theory to achieve deeper insight into the impact of bacterial swimming on vertex fluctuations and reorganization. This point has been partly addressed. In particular, the authors should add a histogram to support their estimate of 10-100 bacteria per compartment ().

We have added a histogram of bacteria per compartment (new Supplementary Figure 10). Based on this histogram we have provided a better estimate of 0 – 40 bacteria per compartment and reference Fig. S10 on page 9 of the revised manuscript.

In addition, the newly added section in which the influence of the bacterial occupancy is discussed (page 9 and Fig. 3c) is not entirely clear. Fig. 3c appears to be based on experimental data from three samples, but it is not clear how the volume fraction was estimated, how it was varied between 0.01 and 0.04 (is this a poisson distribution), which data points correspond to which experiment.

We assume that the referee is discussing Fig. 3d, not Fig. 3c, which is only a cartoon image. Fig. 3d indeed shows experimental results from three samples. We have now colored the repeats to more clearly visualize the statistical power. The volume fraction of each

compartment was not varied systematically; instead, the rather large stochastic variation in the filling of different compartments was exploited (see also previous point). We have explained this approach to sample the statistical variation of bacterial occupancy now more clearly in the main text (page 9).

Moreover, the text suggests this figure shows data for equal loading of the compartments (“We were initially surprised to find that in this model, equal loading of the three compartments with an equal number of swimmers did not enhance vertex fluctuations on average (Fig. 3c).”). However, the figure does not show the average vertex fluctuations, but the standard deviation.

We thank the reviewer very much for identifying an error in our comparison between theoretical and experimental results, where we mistakenly compared the experimentally measured standard deviation with the mean value calculated from the theory. This error has now been corrected, and the updated data (mean vertex displacement measured from experiment) are presented in Fig. 3d. This correction has altered the linear fit shown in the figure panel, which changes the estimate of the swimmer size R from 1.6 to 2 μm . We have previously stated that the size R should be in the range of a individual bacteria. Consequently, our main conclusion that the model adequately fits the experimental data remains unchanged. We have also taken this opportunity to rename the symbols used in the theory section and Supplementary Methods to clearly distinguish them from the symbols used in the experimental data, to avoid further confusion.

Moreover, in their response to point 4, the authors write that “the differential filling of neighboring compartments is the dominant effect on vertex fluctuations.” Does Fig. 3c show data for equal filling or differential filling, and if so, how is the effective volume fraction defined?

The cartoon shows three compartments. In the example, two compartments (green color) are supposed to exhibit equal filling, while the third (white background) shows no filling. We have updated the cartoon in Fig. 3c to indicate the symbols associated Φ_n with each compartment. Values of $\Phi = |\Phi_1 + \Phi_2 + \Phi_3|$ that are close to 0 correspond to equal filling, while higher values indicate increasing degrees of differential filling.

The authors should revise the text and figure presentation and caption to clarify these points.

We hope that revised manuscript now fully addresses the important points raised by the referee.

Additional technical points and writing style:

1. Mix up of Span-80, span-80 and SPAN 80.

We now use Span-80 throughout the text.

2. vertexes/vertices both used

We now use ‘vertices’ consistently in the main text.

3. Inconsistent references to Supplementary (Supplement Fig. 4, Supplementary Fig. S3, Supplementary Fig. 2).

We have addressed the inconsistencies in the references.

4. Movie 3 does not appear to show the T1 transition, but the text somehow suggests it does: it is referenced together with Fig. 2c, which depicts a T1 transition. It would be advisable to include a video from which the snapshots in Fig. 2c were taken.

We have referenced Fig 2c now at the correct location of the main text. For the data shown in Fig. 2c we do not have movie that has better time resolution than the snapshots shown.

5. Typo: “repats” (caption of Fig. 2). Additionally, are these actually repeats? One curve seems to correspond to active and one to a passive system. The caption for Fig. 2e lacks information about the difference between the fits of the two series, and a reference to eq.2 for the fit is incorrect, this should be eq. 1 in Supplementary Note 2.

We thank the referee for allowing us to clarify that for each active and passive condition $n = 2$ repeats were performed. The power spectrum was calculated from the aggregated data of both repeats. We have included the correct reference.

6. Typo: swimming speeds the bacteria (“of” missing)

The typo was fixed on page 7.

7. Fig. 3a: frequency should have a unit.

Fig 3a y-axis was now changed to the correct label “Count”.

8. Fig. 3b: y-axis lacks units. The meaning of the different symbols is not defined. Are these prototissues with and without bacteria?

The referee is correct that this information was missing. The meaning of the symbols was indeed the cases of active/heat-killed, and the correct legend is now included.

9. Fig. 3f: a legend to correlate circle size with the actual displacement magnitude (number) is missing.

We have added the legend for the circle size in the update figure on page 11.